# Consistent Meta-Reinforcement Learning via Model Identification and Experience Relabeling

## Abstract

Reinforcement learning algorithms can acquire policies for complex tasks automatically, however the number of samples required to learn a diverse set of skills can be prohibitively large. While meta-reinforcement learning has enabled agents to leverage prior experience to adapt quickly to new tasks, the performance of these methods depends crucially on how close the new task is to the previously experienced tasks. Current approaches are either not able to extrapolate well, or can do so at the expense of requiring extremely large amounts of data due to on-policy training. In this work, we present *model identification and experience relabeling* (MIER), a meta-reinforcement learning algorithm that is both efficient and extrapolates well when faced with out-of-distribution tasks at test time based on a simple insight: we recognize that dynamics models can be adapted efficiently and consistently with off-policy data, even if policies and value functions cannot. These dynamics models can then be used to continue training policies for out-of-distribution tasks without using meta-reinforcement learning at all, by generating synthetic experience for the new task.

## 1 Introduction

Meta-reinforcement learning algorithms can enable acquisition of new tasks from just a small number of samples by leveraging experience from previous related tasks (Duan et al., 2016; Wang et al., 2016; Finn et al., 2017). However, the performance of these methods on new tasks depends crucially on how close the tasks are to the meta-training task distribution. Meta-learned models can adapt quickly to tasks that are similar to those seen during training, but can lose much of their benefit when adapting to tasks that are too far away from the meta-training distribution. This places a significant burden on the user to carefully construct meta-training task distributions that sufficiently cover the kinds of tasks that may be encountered at test time.

Many meta-RL methods either utilize a variant of model-agnostic meta-learning (MAML) (Finn et al., 2017; Rothfuss et al., 2018; Nagabandi et al., 2018), or an inference-based formulation with recurrent (Duan et al., 2016; Wang et al., 2016), attentional (Mishra et al., 2017), or variational (Rakelly et al., 2019) methods. The latter class of methods generally fails to handle out-of-distribution tasks, because the model cannot adequately deal with out-of-distribution inputs corresponding to the new task. Most of the former class of methods, where gradient-based adaptation corresponds to a well-defined and consistent learning process, require on-policy samples, resulting in high sample complexity for meta-training in order to produce efficient adaptation. In this paper, we make use of a simple insight to develop *model identification and experience relabeling* (MIER), a meta RL algorithm that is both efficient and which extrapolates effectively when faced with out-of-distribution tasks at meta-test time: we recognize that *dynamics and reward models* can be adapted consistently, using MAML-style update rules with off-policy data, even if policies and value functions cannot. These models can then be used to train new policies for out-of-distribution tasks without using meta-reinforcement learning at all, by generating synthetic experience for the new task.

To be able to quickly learn dynamics models, we reformulate the meta-reinforcement learning problem as one of MDP *identification*. We use this approach since identifying a task involves determining its transition dynamics and reward function, which is exactly the information that a model of the

task needs to represent. Specifically, we use a *supervised* meta-learning method that optimizes for a dynamics model initialization such that conditioned on a *MDP context descriptor*, prediction error on a validation batch of data sampled from the task is minimized, after updating only the context descriptor with a few gradient steps. Effective model training requires the validation batch to contain data corresponding to optimal behavior for the tasks, which we obtain by training a universal policy conditioned on the context descriptor. Note that since our formulation ensures that the context descriptor contains sufficient information about the task, the policy does *not* need to be meta-trained or adapted, and can hence be learned with simple and efficient off-policy RL algorithms, without needing to handle the complexity of meta-reinforcement learning.

At test time, given out-of-distribution tasks, the adapted context descriptor would indeed be out of distribution and thus our context-conditioned model and policy might not perform well for the test task. However, since our method uses gradient descent which is a consistent learning method for adaptation, we can continue to improve our model using more gradient updates. We then leverage all of the experience collected from other tasks during meta-training, by using the learned model to relabel the next state and reward information, thus obtaining synthetic data to continue training the policy.

Our main contribution is an off-policy meta-RL algorithm that is sample efficient, stable and which extrapolates well to out-of-distribution tasks. By formulating the meta-adaptation as MDP *identification*, we are able to transform the meta-RL problem into a supervised meta-learning problem and thus benefit from the stability and consistency of supervised learning methods. The consistency of our model also enables us to continue improving our policy without collecting extra data by relabeling data collected from other tasks, thus allowing us to efficiently adapt to out-of-distribution tasks.

## 2 RELATED WORK

Meta-reinforcement learning algorithms extend the framework of meta-learning (Schmidhuber, 1987; Thrun & Pratt, 1998; Naik & Mammone, 1992) to the reinforcement learning setting. Current meta-reinforcement learning algorithms can be roughly categorized into 3 large categories, model-free context based methods, model-free gradient based methods and model based methods.

Model-free, context based methods such as Duan et al. (2016); Wang et al. (2016); Mishra et al. (2017); Rakelly et al. (2019); Humplik et al. (2019) often encode the experience during adaptation into a latent context variable, and the policy is conditioned on the task specific context to adapt to a given task. The context encoding process is often done via a recurrent network (Duan et al., 2016; Wang et al., 2016), via an attention mechanism (Mishra et al., 2017) or via amortized probabilistic inference (Rakelly et al., 2019; Humplik et al., 2019). While many of these methods can efficiently summarize experiences during adaptation, it is often hard for them to encode any universal learning algorithm such as gradient descent. This is because given a fixed distribution of training tasks, the most efficient way of adapting to a particular task is to directly infer it from the task distribution, rather than applying any universal learning algorithm. While efficient in handling in distribution tasks, the lack of universal learning algorithm in these methods implies that they would be able to adapt to out of distribution tasks. Our method, on the other hand, is able to handle out of distribution tasks through continual adaptation with gradient descent.

Model-free gradient based meta-RL methods, such as Finn et al. (2017); Rothfuss et al. (2018); Zintgraf et al. (2018); Rusu et al. (2018); Liu et al. (2019), implement gradient descent as the adaptation process. The parameters are often optimized such that the model achieves good performance after only a few steps of gradient descent. Since gradient descent is a universal learning algorithm, with a large capacity model, these methods are consistent in the way that they would guarantee to improve even on out-of-distribution tasks. However, most of these methods are based on on-policy RL algorithm, which means they are rather sample inefficient at training time and requires more data to be collected when adapting to new tasks. It is worth noting that there are also works that combine gradient based and context based methods such as Lan et al. (2019). However such methods still suffer from the same sample efficiency problem as other gradient based methods because of the use of on-policy gradient descent. Our method mitigate this problem by applying an off-policy RL algorithm on top of a gradient based meta-learning algorithm to achieve better sample efficiency at training time. At test time, our method reuses the experiences collected during training to enable fast adaptation without collecting a lot more data.

Model based meta-RL methods such as Nagabandi et al. (2018); Sæmundsson et al. (2018), a rapidly adapting model instead of a policy is learned. Since learning a fast adapting model is a supervised meta-learning problem, a broader range of meta-learning algorithms can be applied. If an universal learning algorithm is used to represent the adaptation process, the model learning can then be made consistent such as in Nagabandi et al. (2018). At test time, when the model is adapted to a particular task, standard planning techniques such as model predictive control is often applied to select actions. While possible to be consistent, the model based meta-RL methods often suffer in long horizon tasks due to the accumulated error of model while planning long time ahead into the future. Our method does not suffer from this problem since we only use one step prediction of our model to train a model-free policy, and only execute the policy during test time. This approach is similar to methods presented in Sutton (1991); Janner et al. (2019).

## 3 PRELIMINARIES

### 3.1 REINFORCEMENT LEARNING

Formally, a reinforcement learning problem is defined by a Markov decision process (MDP). We adopt the standard definition of a MDP $\mathcal{T} = (\mathcal{S}, \mathcal{A}, p, \mu_0, r, \gamma)$, where $\mathcal{S}$ is the state space, $\mathcal{A}$ is the action space, $p(\mathbf{s}'|\mathbf{s}, \mathbf{a})$ is the unknown transition probability of landing on next state $\mathbf{s}'$ at the next time step when an agent takes action $\mathbf{a}$ at state $\mathbf{s}$, $\mu_0(\mathbf{s})$ is the initial state distribution, $r(\mathbf{s}, \mathbf{a})$ is the reward function, and $\gamma \in (0, 1)$ is the discount factor. An agent acts according to some policy $\pi(\mathbf{a}|\mathbf{s})$ and the objective of learning is to maximize the expected rewards $\mathbb{E}_{\mathbf{s}_t, \mathbf{a}_t \sim \pi}[\sum_t \gamma^t r(\mathbf{s}_t, \mathbf{a}_t)]$.

### 3.2 MODEL BASED REINFORCEMENT LEARNING

In model based reinforcement learning, a model $\hat{p}(\mathbf{s}', \mathbf{r}|\mathbf{s}, \mathbf{a})$ that predicts the reward and next state from current state and action is trained using standard supervised learning approaches. The model is then used to generate data to train a policy. Specifically, given a model, we perform the following optimization to obtain the policy: $\arg\max_\pi \mathbb{E}_{\mathbf{s}_t, \mathbf{a}_t \sim \pi, \hat{p}}[\sum_t r(\mathbf{s}_t, \mathbf{a}_t)]$. Note here the expectation is taken with respect to the model distribution of states instead of the true environment distribution.

### 3.3 META-REINFORCEMENT LEARNING

In meta-reinforcement learning, we represent various of similar tasks with a distribution of MDPs $\rho(\mathcal{T})$, where each specific task is a sample drawn from the distribution. Given a specific task $\mathcal{T}$, the agent is allowed to collect little amount of data $\mathcal{D}_{adapt}^{(\mathcal{T})}$, adapt the policy to form $\pi_{\mathcal{T}}$ according to the data. The objective of this problem is to maximize the expected rewards of the adapted policy $\mathbb{E}_{\mathcal{T} \sim \rho(\mathcal{T}), \mathbf{s}_t, \mathbf{a}_t \sim \pi_{\mathcal{T}}}[\sum_t \gamma^t r(\mathbf{s}_t, \mathbf{a}_t)]$.

### 3.4 SUPERVISED META-LEARNING AND MODEL AGNOSTIC META LEARNING

We briefly introduce the supervised meta-learning problem and the model agnostic meta-learning approach, which is an important foundation of our work. In supervised meta-learning, we also have a distribution of tasks $\rho(\mathcal{T})$ similar to the meta-RL setup, except that the task $\mathcal{T}$ is now a pair of input and output random variables $(X_{\mathcal{T}}, Y_{\mathcal{T}})$. Given a small dataset $\mathcal{D}_{adapt}^{(\mathcal{T})}$ sampled from a specific task $\mathcal{T}$, the objective is to built a model that performs well on the evaluation data $\mathcal{D}_{eval}^{(\mathcal{T})}$ sampled from the same task. If we denote our model as $f(X; \theta)$, the adaptation process as $\Xi(\theta, \mathcal{D}_{adapt}^{(\mathcal{T})})$ and our loss function as $\mathcal{L}$, the objective can be written as:

$$\min_{f, \Xi} \mathbb{E}_{\mathcal{T} \sim \rho(\mathcal{T})}[\mathcal{L}(f(X_{\mathcal{T}}; \Xi(\theta, \mathcal{D}_{adapt}^{(\mathcal{T})})), Y_{\mathcal{T}})]$$

Model agnostic meta-learning (Finn et al., 2017) is an approach to solve the supervised meta-learning problem. Specifically, the model $f(X; \theta)$ is represented as a neural network, and the adaptation process is represented as few steps of gradient descent. For simplicity of notation, we only write out the one step of gradient descent:

$$\Xi_{\text{MAML}}(\theta, \mathcal{D}_{adapt}^{(\mathcal{T})})) = \theta - \alpha \nabla_\theta \mathbb{E}_{X, Y \sim \mathcal{D}_{adapt}^{(\mathcal{T})}}[\mathcal{L}(f(X; \theta), Y)]$$

Note that because $\Xi_{\text{MAML}}$ is the standard gradient descent, it does not contain any trainable parameters and hence the trainable parameters are all contained in the network weights $\theta$. Therefore, the training process of MAML can be summarized as optimizing the loss of the model after few steps of gradient descent on data from the new task.

## 4 CONSISTENT META-OPTIMIZATION BY RELABELING EXPERIENCE

In this section, we lay the meta-training and test time adaptation process of MIER. We first reformulate the meta-RL problem into a meta-model identification problem, where we train a fast adapting model to rapidly identify transition dynamics and reward function for a given new task. We parameterize the model with a latent context descriptor to contain all task specific information aquired during adaptation. We then train a universal policy that condition on context descriptor to solve all tasks sampled from the training distribution. At test time, given a potentially out of distribution task, we continual to improve our policy by using our model to relabel data from previously experienced tasks. The consistency of our model adaptation process and policy improvement ensures that the policy will continue to improve even the task is out-of-distribution. We provide the detailed explain of each part of our method in the following sections,

---

**Algorithm 1** Meta-Training of MIER

**INPUT:** task distribution $\rho(\mathcal{T})$, model training steps $N_{model}$, policy training steps $N_{policy}$, learning rate $\alpha$
**OUTPUT:** policy parameter $\psi$, model parameter $\theta$, model context $\phi$

Randomly initialize policy parameter $\psi$, model parameter $\theta$, model context $\phi$
Initialize multitask replay buffer $\mathcal{R}(\mathcal{T}) \leftarrow \emptyset$
**while** $\theta, \phi, \psi$ *not converged* **do**
    Sample task $\mathcal{T} \sim \rho(\mathcal{T})$
    Collect data batch $\mathcal{D}_{adapt}^{(\mathcal{T})}$ before adaptation using $\pi_\psi$ and $\phi$
    Compute adapted context $\phi_\mathcal{T} = \Xi_{\text{MAML}}(\theta, \phi, \mathcal{D}_{adapt}^{(\mathcal{T})})$
    Collect data batch $\mathcal{D}_{eval}^{(\mathcal{T})}$ after adaptation using $\pi$ and $\phi_\mathcal{T}$
    Add data to replay buffer $\mathcal{R}(\mathcal{T}) \leftarrow \mathcal{R}(\mathcal{T}) \cup \mathcal{D}_{adapt}^{(\mathcal{T})} \cup \mathcal{D}_{eval}^{(\mathcal{T})}$
    **for** $i = 1$ *to* $N_{model}$ **do**
        Sample task in replay buffer $\mathcal{T} \sim \mathcal{R}$
        Sample two batches of data $\mathcal{D}_{adapt}^{(\mathcal{T})}, \mathcal{D}_{eval}^{(\mathcal{T})} \sim \mathcal{R}(\mathcal{T})$
        Train model $(\theta, \phi) \leftarrow (\theta, \phi) - \alpha \nabla_{\theta, \phi} J_{\hat{p}}(\theta, \phi, \mathcal{D}_{adapt}^{(\mathcal{T})}, \mathcal{D}_{eval}^{(\mathcal{T})})$
    **end**
    **for** $i = 1$ *to* $N_{policy}$ **do**
        Sample task in replay buffer $\mathcal{T} \sim \mathcal{R}$
        Sample two batches of data $\mathcal{D}_{adapt}^{(\mathcal{T})}, \mathcal{D}_{eval}^{(\mathcal{T})} \sim \mathcal{R}(\mathcal{T})$
        Compute adapted context $\phi_\mathcal{T} = \Xi_{\text{MAML}}(\theta, \phi, \mathcal{D}_{adapt}^{(\mathcal{T})})$
        Train policy $\psi \leftarrow \psi - \alpha \nabla_\psi J_\pi(\psi, \mathcal{D}_{eval}^{(\mathcal{T})})$
    **end**
**end**

---

and we lay out the pseudo code for meta-training time in Algorithm 1 and test time in Algorithm 2.

### 4.1 META-MODEL IDENTIFICATION WITH LATENT CONTEXT

In a meta-RL problem, tasks are drawn from a distribution of MDPs, and an agent adapts to each of the tasks by observing some data for the tasks. The adaptation process is usually performed as either an implicit or explicit inference of some task representations. That is, the agent adapts to a task by identifying it from other tasks in the same distribution. In a distribution of MDPs, the only varying factors are the dynamics $p$ and the reward function $r$. Therefore, a sufficient condition for identifying the task is to learn the transition dynamics and the reward function, and this is exactly what model based RL methods do. Hence, we can naturally formulate the meta-task identification problem as model based RL and solve it with supervised meta-learning methods.

Specifically, we choose the MAML method for its simplicity and consistency. Unlike the standard supervised MAML formulation, we condition our model on a latent context vector, and we only change the context vector when adapting to new tasks. With this formulation, we restrict all the task specific information learned during adaptation to the context vector, and thus allowing the policy to condition on it. Let us denote our model as $\hat{p}(\mathbf{s}', \mathbf{r}|\mathbf{s}, \mathbf{a}; \theta, \phi)$, where $\theta$ is the neural network parameters and $\phi$ is the latent context vector that is passed in as input to the network. One step of gradient adaptation can be written as follows:

$$\phi_\mathcal{T} = \Xi_{\text{MAML}}(\theta, \phi, \mathcal{D}_{adapt}^{(\mathcal{T})})) = \phi - \alpha \nabla_\phi \mathbb{E}_{(\mathbf{s}, \mathbf{a}, \mathbf{s}', \mathbf{r}) \sim \mathcal{D}_{adapt}^{(\mathcal{T})}} [-\log \hat{p}(\mathbf{s}', \mathbf{r}|\mathbf{s}, \mathbf{a}; \theta, \phi)]$$

We use the log likelihood as our objective for the probabilistic model. Then we evaluate the adapted context vector $\phi_{\mathcal{T}}$ and minimize its loss on the evaluation dataset to learn the model. Specifically, we minimize the model meta-loss function $J_{\hat{p}}(\theta, \phi, \mathcal{D}_{adapt}^{(\mathcal{T})}, \mathcal{D}_{eval}^{(\mathcal{T})})$ to obtain the optimal parameter $\theta$ and context vector initialization $\phi$.

$$\arg\min_{\theta, \phi} J_{\hat{p}}(\theta, \phi, \mathcal{D}_{adapt}^{(\mathcal{T})}, \mathcal{D}_{eval}^{(\mathcal{T})}) = \arg\min_{\theta, \phi} \mathbb{E}_{(\mathbf{s}, \mathbf{a}, \mathbf{s}', \mathbf{r}) \sim \mathcal{D}_{eval}^{(\mathcal{T})}} [-\log \hat{p}(\mathbf{s}', \mathbf{r} | \mathbf{s}, \mathbf{a}; \theta, \phi_{\mathcal{T}})]$$

The main difference between our method and previously proposed context based meta-RL methods (Rakelly et al., 2019; Duan et al., 2016) is that we use gradient descent to adapt the context. Therefore, given a model with large enough capacity, the adaptation process is consistent, in the sense that it will continue to improve regardless of which task it is presented with. In-distribution tasks will of course have much faster adaptation, since the meta-training process explicitly optimizes for this objective, but the model for out-of-distribution tasks will still adapt to the task given enough samples and gradient steps, since the adaptation process corresponds to a well-defined and convergent learning process.

## 4.2 POLICY OPTIMIZATION WITH LATENT CONTEXT

Given the latent context variable from the adapted model, the meta-RL problem can be effectively reduced to a standard RL problem, as the task specific information has been all encoded in the context variable. Therefore we can apply any model-free RL algorithm to obtain a policy, as long as we condition the policy on the latent MDP descriptor context.

Specifically, we choose the soft actor critic (SAC) method (Haarnoja et al., 2018) for its sample efficiency and stability. Let us parameterize our policy $\pi_\psi$ by a parameter $\psi$. The soft actor critic method maintains an estimate of the values for the current policy $Q^{\pi_\psi}(\mathbf{s}, \mathbf{a}, \phi_{\mathcal{T}}) = \mathbb{E}_{\mathbf{s}_t, \mathbf{a}_t \sim \pi_\psi}[\sum_t \gamma^t r(\mathbf{s}_t, \mathbf{a}_t) | \mathbf{s}_0 = s, \mathbf{a}_0 = a, \mathcal{T}]$ and $V^{\pi_\psi}(\mathbf{s}, \phi_{\mathcal{T}}) = \mathbb{E}_{\mathbf{s}_t, \mathbf{a}_t \sim \pi_\psi}[\sum_t \gamma^t r(\mathbf{s}_t, \mathbf{a}_t) | \mathbf{s}_0 = s, \mathcal{T}]$ through Bellman backup, and improves the policy by minimizing the KL divergence between the policy and the exponential advantage $J_\pi(\psi, \mathcal{D}) = \mathbb{E}_{\mathcal{D}}[D_{KL}(\pi_\psi || \exp\{Q^{\pi_\psi} - V^{\pi_\psi}\})]$ over some dataset $\mathcal{D}$. Note that we condition our value functions $Q^{\pi_\psi}$, $V^{\pi_\psi}$ and policy $\pi_\psi$ all on the adapted task specific context vector $\phi_{\mathcal{T}}$ so that the policy and value functions can access full information about the task.

## 4.3 CONTINUAL ADAPTATION AND EXTRAPOLATION THROUGH DATA RELABELING

At meta-test time, when the model is given an unseen task $\mathcal{T}$, it will first sample a small batch of data and obtain the latent context $\phi_{\mathcal{T}}$ by running the gradient descent adaptation process on the context variable. If the new task is in-distribution, we can directly feed $\phi_{\mathcal{T}}$ to the policy. However, in many cases, we might encounter tasks that are out of distribution. If the task $\mathcal{T}$ is out of the meta-training distribution, then the obtained latent context $\phi_{\mathcal{T}}$ will also be out of distribution for the policy. Even though our model is adapted with a consistent method, the improvement guarantee is only for the model – the latent context $\phi_{\mathcal{T}}$ that might be obtained for an out-of-distribution task might not immediately produce an effective policy. However, with an improved model, we can generate new

---

**Algorithm 2** Test Time Adaptation of MIER

**INPUT:** test task $\hat{\mathcal{T}}$, multitask replay buffer $\mathcal{R}(\mathcal{T})$, model training steps $N_{model}$, policy training steps $N_{policy}$, policy parameter $\psi$, model parameter $\theta$, model context $\phi$, learning rate $\alpha$
**OUTPUT:** policy parameter $\psi$, model parameter $\theta$, model context $\phi$

Collect data batch $\mathcal{D}_{adapt}^{(\hat{\mathcal{T}})}$ from test task $\hat{\mathcal{T}}$ using $\pi_\psi$ and $\phi$

Compute adapted context $\phi_{\hat{\mathcal{T}}} = \Xi_{\text{MAML}}(\theta, \phi, \mathcal{D}_{adapt}^{(\hat{\mathcal{T}})})$

**while** $\theta, \psi$ *not converged* **do**
  **for** $i = 1$ *to* $N_{model}$ **do**
    Train model $\theta \leftarrow \theta - \alpha \nabla_\theta \mathbb{E}_{\mathcal{D}_{adapt}^{(\hat{\mathcal{T}})}}[-\log \hat{p}(\mathbf{s}', \mathbf{r} | \mathbf{s}, \mathbf{a}; \theta, \phi_{\hat{\mathcal{T}}})]$
  **end**
  **for** $i = 1$ *to* $N_{policy}$ **do**
    Sample task in replay buffer $\mathcal{T} \sim \mathcal{R}$
    Sample data batch $\mathcal{D}^{(\mathcal{T})} \sim \mathcal{R}(\mathcal{T})$
    Relabel data $\hat{\mathcal{D}}^{(\hat{\mathcal{T}})} \leftarrow \textbf{Relabel}(\mathcal{D}^{(\mathcal{T})}, \theta, \phi_{\hat{\mathcal{T}}})$
    Train policy $\psi \leftarrow \psi - \alpha \nabla_\psi J_\pi(\psi, \hat{\mathcal{D}}^{(\hat{\mathcal{T}})})$
  **end**
**end**

---

data for the task without sampling from the environment to continue training the policy, providing a sample-efficient (but computationally more costly) adaptation method for out-of-distribution tasks.

When using data generated from a learned model to train a policy, a common caveat is that the model's predicted trajectory often diverges from the trajectory under the real dynamics in long horizon tasks, due to accumulated error. In the special case of meta-RL, we argue that this issue can be avoided by the multi-task setup in meta-RL. Although we have never seen this particular new task, we have seen many other tasks that share the same state space and action space, and we have collected a lot of experience from these tasks. Using the adapted model and policy, we can *relabel* the transitions $(\mathbf{s}, \mathbf{a}, \mathbf{s}', \mathbf{r})$ from any other tasks by sampling a new actions with our adapted policy, and sample next states and rewards with the adapted model. The relabeling process can be written as $\mathbf{Relabel}(\mathcal{D}, \theta, \phi_{\mathcal{T}}) = \{(\mathbf{s}, \mathbf{a}, \mathbf{s}', \mathbf{r}) | (\mathbf{s}, \mathbf{a}) \in \mathcal{D}; (\mathbf{s}', \mathbf{r}) \sim \hat{p}(\mathbf{s}', \mathbf{r} | \mathbf{s}, \mathbf{a}; \theta, \phi_{\mathcal{T}})\}$. We use these relabeled transitions to continue training the policy. By doing so, we effectively transferred experiences from other tasks into this new task, and thus avoided the divergence problem of sampling long horizon trajectories as we are only doing one step prediction. This relabeling scheme is similar to the Dyna algorithm introduced by Sutton (1991). By relabeling experiences from tasks seen before, we can efficiently reuse the large amount of data collected from other tasks to improve our policy on this new task.

## 5 EXPERIMENTAL EVALUATION

We aim to answer the following questions in our experiments: (1) Can MIER meta-train efficiently on standard meta-RL benchmarks without requiring large amounts of data? (2) How does MIER compare to prior meta-learning approaches for extrapolation to meta-test tasks with out-of-distribution (a) reward functions and (b) dynamics? (3) How important is cross task relabeling in leveraging the model to learn an effective policy efficiently?

To answer these questions, we first compare the sample efficiency of MIER to existing methods on several standard meta-RL benchmark environments. We then test MIER on a set of environment with out of distribution meta-test tasks to analyze extrapolation performance. We further run an ablation of our method without cross task labelling, and show that this isn't able to adapt the policy as effectively as our method.

### 5.1 SAMPLE EFFICIENCY ON META-RL BENCHMARKS

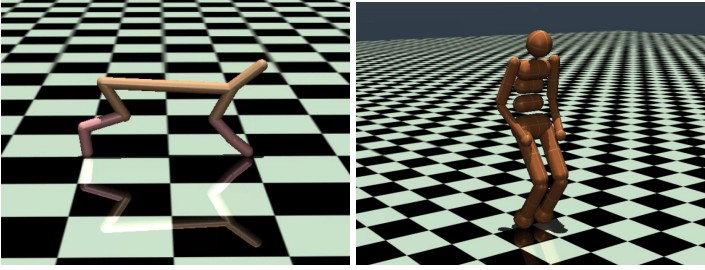

Figure 1: HalfCheetah and Humanoid robot simulated in MuJoCo physics engine. Our experiments use these two domains.

We evaluate MIER on two standard meta-RL benchmark environments based on the MuJoCo physics simulator (Todorov et al., 2012), shown in Figure 1. These environments were introduced by Finn et al. (2017), and later used by the state-of-the-art context based meta-RL method Rakelly et al. (2019). They include half cheetah robot running at different velocities (HalfCheetah-Vel) and a humanoid robot running in different directions (Humanoid-Dir). For all environments, we follow the exact setting in Rakelly et al. (2019). For comparison, we include performance of the PEARL algorithm (Rakelly et al., 2019) and MAML algorithm (Finn et al., 2017) as our baselines, and we average the performance of all algorithms across 3 random seeds.

We plot the test time performance after adaptation versus number of steps in the environment in Figure 2. We see that our method is able to achieve performance comparable to the state of the art

method in meta-RL in terms of sample efficiency (PEARL). This is due to the ability of our method to perform supervised meta-learning for the model, and off-policy reinforcement learning for the policy, which results in a stable and efficient optimization procedure during meta-training. Note that here, off-policy refers to the *meta-training* process. However, as we discuss in Section 4.3, our method can also perform off-policy *adaptation* via cross task relabeling. We evaluate this in the next section.

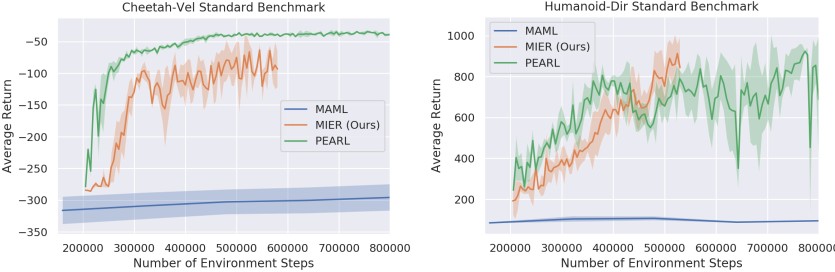

Figure 2: Performance on standard meta-RL benchmarks for **left:** Half Cheetah velocity **right:** Humanoid direction. The return is evaluated on (in-distribution) meta-test tasks at different steps during the *meta-training* process, following the protocol from Rakelly et al. (2019). Note that both our method and PEARL are substantially more efficient than the on-policy MAML algorithm, and our method performs comparably to the off-policy PEARL algorithm.

## 5.2 EXTRAPOLATION TO OUT-OF-DISTRIBUTION TASKS

We evaluate the extrapolation capability of MIER separately on environments with varying reward functions, and those with different dynamics. In order to ensure fair comparisons, we train all algorithms to convergence before testing on new tasks. Hence, MAML is trained for 80 millions steps, while PEARL and MIER only require about a million steps. Unless stated otherwise, we adapt with MIER using cross task relabeling in all of these experiments.

### 5.2.1 ADAPTATION TO OUT-OF-DISTRIBUTION REWARDS

For testing extrapolation ability to tasks with different rewards, we take the HalfCheetah-Vel environment from the standard benchmarks and restrict the meta-training task distribution. This is required since, in the original environments, the meta-training distribution covers almost the entire feasible task space, leaving little room for extrapolation. Instead, if we limit the training distribution, we can easily find out-of-distribution tasks to test the extrapolation capability of our method.

The training tasks consist of target velocities at which the cheetah needs to run sampled uniformly from 0 to 1. We then evaluate the ability of the meta-learned polices to adapt in order to run at new speeds chosen at 0.5 intervals from 0.5 to 2.5, using only 2 trajectories sampled from the environment. From Figure 3, we see that MIER attains much higher return than other methods for target velocities that are substantially outside of the meta-training distribution. Eventually, performance degrades for all methods, since the state space distribution differs too much from the meta-training tasks, making cross task relabeling ineffective. Nonetheless, this experiment shows that MIER's capacity to adapt to out-of-distribution tasks substantially exceeds that of prior meta-RL methods.

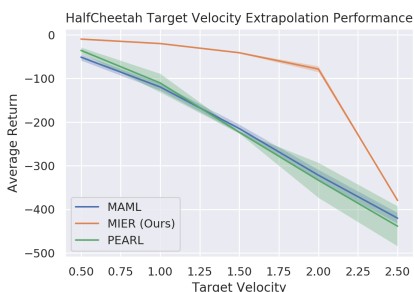

Figure 3: Target velocity (reward function) extrapolation for HalfCheetah environment.

### 5.2.2 ADAPTATION TO OUT-OF-DISTRIBUTION DYNAMICS

To study adaptation to out-of-distribution dynamics, we take the HalfCheetah environment, and randomly negate the control of 3 joints out of the total 6 joints of the robot. During meta-training, we only select joints from the first 5 joints to negate, and during meta-test time, we select 2 joints from the first 5 and always negate the 6th joint. This ensures that the agent never sees the negation of the

6th joint during meta-training time, and hence needs to extrapolate well to get good return on the test tasks. In total, we have 10 training tasks and 10 test tasks.

We compare the return of all methods after adapting on 2 new trajectories of the validation task. From the plot of average performance across test tasks, in Figure 4, we see that MIER achieves superior performance compared to existing methods. We are able to extrapolate so efficiently since we are able to leverage off-policy data collected during meta-training. This in turn is enabled by our ability to learn a model quickly using minimal data, which is the consistent meta-training objective.

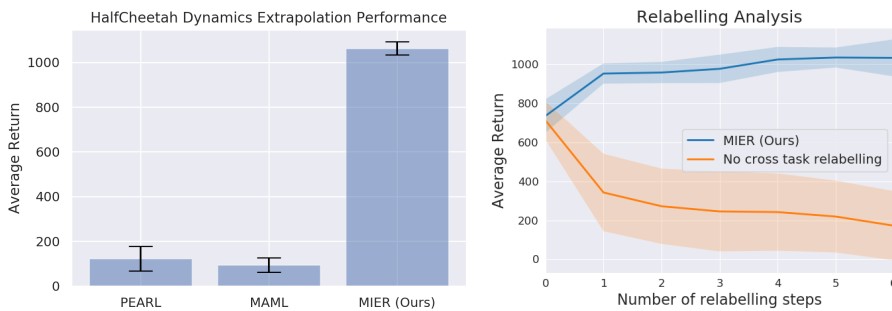

Figure 4: Dynamics extrapolation on HalfCheetah with negated joints. **Left:** Return after adaptation for each method. **Right:** Ablation analysis on the effect of relabeling. The results show that MIER with cross task relabeling substantially outperforms prior methods when adapting to this task with out-of-distribution dynamics. The ablation analysis shows that the cross-task relabeling is essential to attain this level of performance.

To further highlight the importance of relabeling data from other tasks during adaptation, we run an ablation where the policy is learned only using the sampled data. Figure 4 shows performance of the policy plotted against the number of steps where we relabel exsiting data and continue to train the policy. We see that without the benefit of off-policy data, the policy performance deteriorates rapidly due to overfitting to the small batch of data sampled from the test task.

## 6    CONCLUSION

In this paper, we introduce a consistent and sample efficient meta-RL algorithm by reformulating the meta-RL problem as model identification. Our algorithm can adapt to new tasks by determining the parameters of the model, which predicts the reward and future transitions. This allows us to perform meta-learning via supervised learning of the model, which is more stable and sample efficient. More importantly, this provides us with a *consistent* adaptation procedure, where adaptation is performed via gradient descent. This means that, even for out-of-distribution meta-test tasks, our method eventually learns the right model. This model then allows us to *relabel* past experience to reuse it for the new task, running off-policy reinforcement learning to acquire the policy for the new task. Experiment results show that our method achieves superior performance compared to existing methods, especially on out-of-distribution tasks that requires extrapolation, where consistent adaptation of the model followed by relabeling and off-policy reinforcement learning substantially outperforms prior methods that adapt to the new task directly.

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

# Appendices

## A   IMPLEMENTATION DETAILS

The implementation of the algorithm largely builds on the Model Based Policy Optimization code ( Janner et al. (2019)), and the environments are standard meta-RL benchmarks, with slight modifications for the extrapolation tasks as described in the experiment section. The context vector is a variable that is appended to the states and actions when sent to the model, and appended to the states when passed to the policy. The hyper-parameters are kept mostly fixed across all experiments. One major change is that environments with changing dynamics have models predicting both next state and reward, while the environments with only changing reward have prediction just for reward.

Some key hyperparameters (common across experiments):

1. Number of fast adapt steps for model meta-training : 2
2. Fast adapt learning rate for the model : 0.1
3. Batch size for model adaptation : 256
4. Reward-dynamics model architecture : 200-200-200-200
5. Dimension of context vector : 5
6. Regression weight on post-update contexts : 0.01
7. Number of tasks sampled per training epoch : 5
8. Number of data points sampled from a task per training epoch : 1000
9. Number of SAC training steps per training epoch : 1000
10. Learning rate for Q functions in SAC : 0.0003
11. Q network hidden layers : 256-256

