# OpenReview forum: "Consistent Meta-Reinforcement Learning via Model Identification and Experience Relabeling"
_ICLR.cc/2020/Conference — Reject_

### Official Review · AnonReviewer1 · 2019-10-11
**Official Blind Review #1**

**Rating:** 6

**Review:**

This paper proposed a novel approach to reformulate the meta-RL problem as model identification with a gradient descent based algorithm. Such innovation not only allowed us to perform meta-learning via supervised learning of the model, which is more stable and sample efficient, but also allowed us to leverage off-policy RL to learn the policy instead of meta RL.

Pros:
1. Paper clarity. Although the submission had a few typos (I am not a native English speaker, but I'd encourage the authors to polish the writing of the paper), it's a very well-written paper overall. The flow and logic of this paper was clean, and the authors stroke a good balance between being focused about the core contribution of the paper, and reviewing related work and introducing sufficient preliminaries. As a result, I think this paper was accessible to both domain experts and the broader ICLR community.

2. Novelty. This paper proposed a novel approach to reformulate the meta-RL problem as model identification with a gradient descent based algorithm. To the best of my knowledge, this was the first paper broke the meta-RL problem into a simpler meta supervised learning problem and an off-policy RL learning problem. Although each component of the proposed solution was not new, e.g., "relabel" was used in Dyna, MAML was first introduced in 2017, the combination of each component to address the meta-RL problem seemed to the novel to me. And I think the idea could be interesting to the ICLR community.


Cons:
It's weak accept rather than accept from me because of how the empirical evaluation were conducted in the paper, and I think the experiments conducted in the paper were a little bit weak (common for most ICLR submissions). Examples:

1. Number of gradient steps is an important tuning parameter for MAML, it would be interesting to discuss number of gradient steps within the context of MIER.
2. It might be useful to conduct some qualitative results to understand the model learned with MIER against the baselines, e.g., how well MIER adapt to the out-of-distribution tasks with simulated data points (examples o such qualitative studies could be found, say, in Finn et al., 2017).
3. Given the fact that one major contribution of this paper was reformulating the meta-RL problem as model identification, it would be useful to conduct some quantitative study to help the readers understand the effectiveness of learning the environment model p(s’, r|s,a) compared to ground-truth, and how the quality of the learned environment model made an impact on the overall performance of the model.
4. Some implementation details of MIER were missing, I don’t feel confident about how reproducible this research would be. For example, the specification of both environment and policy models were not discussed in the paper.
5. In general, it would be useful to conduct more experiment results on more diverse data sets, say, in a supplement material.


A few questions to the authors:
1. Section 3.2: I assume the expectation should be taken w.r.t. p’ rather than f?
2. In Algorithm 1 & 2, how was the adapted context \phi_T used to update policy \psi? Was it as input to the model parametrized by \psi? It might be useful to make it clearer.


**Experience Assessment:**

I have read many papers in this area.

**Review Assessment: Checking Correctness Of Derivations And Theory:**

I assessed the sensibility of the derivations and theory.

**Review Assessment: Checking Correctness Of Experiments:**

I assessed the sensibility of the experiments.

**Review Assessment: Thoroughness In Paper Reading:**

I read the paper thoroughly.

---

> ### Author Response · Authors · 2019-11-14
> **Author Reply for Official Blind Review #1**
>
>
> First we want to thank reviewer #1 for the constructive comments. We have updated the paper to incorporate the reviewer’s suggestions.
>
> Regarding point #1, as suggested by the reviewer, we have conducted experiments for an ablation study of the algorithm performance vs the number of gradient steps in the half cheetah velocity environments, and the results can be found at https://imgur.com/a/e6qUNCH . We see that there is a trade off between performance improvement and stability when increasing the number of fast adaptation steps. We will incorporate these results in the final version of the paper.
>
> Regarding point #2, as suggested by the reviewer, we will modify the paper to include a point mass toy examples and visualization of the policy behavior.
>
> As for point #3, as suggested by the reviewer, we have conducted experiments for an analysis of model prediction error, and the results can be found at https://imgur.com/a/5UdqLyp . We see that the model loss does decrease during training, which matches the improvement in average return. We will incorporate these results in the final version of the paper.
>
> For point #4, we have modified the paper to include all the hyperparameter configurations in Appendix A.
>
> For point #5, as suggested by the reviewer, we have conducted additional in-distribution experiments in the ant-direction environment used in other meta-RL papers, and the results can be found at https://imgur.com/a/F8W37TD . We have also conducted additional experiments for out-of-distribution tasks in the humanoid direction environment, and the results can be found at https://imgur.com/a/ZxRRTmd . The x-axis corresponds to different test tasks, where task 0 is the easiest and -5 and 5 are the most out-of-distribution. We see that for almost all tasks, our methods outperforms PEARL. We will incorporate these results in the final version of the paper.
>
>
> Answers to questions:
> Q1: Section 3.2: I assume the expectation should be taken w.r.t. p’ rather than f?
>
> A1: That’s indeed a typo. Thanks for pointing it out!
>
> Q2: In Algorithm 1 & 2, how was the adapted context \phi_T used to update policy \psi? Was it as input to the model parametrized by \psi? It might be useful to make it clearer.
>
> A2: We have 2 phases of improving the policy with the adapted model context. For the first phase, we direct feed the updated model context as part of the input to the policy. For continual improvement, we use data generated by the model conditioned on the adapted context to continue training the policy.

---

### Official Review · AnonReviewer2 · 2019-10-22
**Official Blind Review #2**

**Rating:** 3

**Review:**

Summary
-------------
The authors propose an algorithm for meta-rl which reduces the problem to one of model identification. The main idea is to meta-train a fast-adapting model of the environment and a shared policy, both conditioned on task-specific context variables. At meta-testing, only the model is adapted using environment data, while the policy simply requires simulated experience. Finally, the authors show experimentally that this procedure better generalizes to out-of-distribution tasks than similar methods.

Major comments
--------------
Making meta-rl algorithm generalize better outside of the meta-training distribution is a relevant open problem, and this work proposes nice ideas towards its solution. The paper is well-organized and easy to read. The idea of reducing meta-rl to a task identification problem is not completely novel since some recent works have been proposed in this direction (see later). Anyway, the proposed approach is interesting and seems (at least from the proposed experiments) effective. My main concerns follow.

1. Though they attempt to address all relevant questions about the proposed approach, I found the experiments quite weak. Only two Mujoco domains are used for the standard meta-rl experiment, and only one of them (HalfCheetah) is used to test the out-of-distribution capabilities. Regarding the first experiment, MIER always performs comparably or worse than PEARL. What is the intuition behind this result? Does it suggest that MIER is paying additional sample complexity in "in-distribution" tasks in order to be more robust to out-of-distribution ones? On the other hand, the generalization experiments seem much more promising, but I would like to see more (at least the humanoid robot as well) to confirm that this result is not only a specific case of this domain. Furthermore, from Figure 3 it seems that MIER improves over PEARL even on in-distribution tasks, while it performed significantly worse in Figure 2. Why does this happen?

2. Related to the previous point, I did not find any description of the parameters adopted in all experiments (learning rates, batch sizes, etc.). I do not believe I would be able to reproduce the results at the present time.

3. The proposed method is somewhat related to other recent works [1,2]. In particular, [2] presents similar ideas, where the authors meta-learn a fast-adapting model (actually, a task encoder) and a shared universal policy conditioned on the task representation. The main focus is still to improve the generalization to out-of-distribution tasks. Can the authors better discuss the relations to these works?

Minor comments
--------------
- In the introduction: "Effective model training requires the validation batch to contain data corresponding to optimal behavior for the tasks...". Why? In principle we could train a good model of the environment by running a sufficiently-explorative policy.
- In the related works: "Our method does not suffer from this problem since we use our model to train a model-free policy". It is not clear why (though it becomes later) since simulating long trajectories from a learned model could lead to the usual divergence issues.
- In Sec. 3.2: r in \hat{p} should not be bold. Also, "f" in the subscript of the expectation was not defined (is it \hat{p}?).
- In Sec 3.4: there is a minimization over \phi which however does not appear in the objective.
- In the optimization problem at page 5: \phi_{\Tau} should probably be \phi.
- In Fig. 2, why is MIER run for less steps than the other algorithms?

[1] Humplik, J., Galashov, A., Hasenclever, L., Ortega, P. A., Teh, Y. W., & Heess, N. (2019). Meta reinforcement learning as task inference. arXiv preprint arXiv:1905.06424.
[2] Lan, L., Li, Z., Guan, X., & Wang, P. (2019). Meta Reinforcement Learning with Task Embedding and Shared Policy. IJCAI 2019.

**Experience Assessment:**

I have published one or two papers in this area.

**Review Assessment: Checking Correctness Of Derivations And Theory:**

I assessed the sensibility of the derivations and theory.

**Review Assessment: Checking Correctness Of Experiments:**

I assessed the sensibility of the experiments.

**Review Assessment: Thoroughness In Paper Reading:**

I read the paper thoroughly.

---

> ### Author Response · Authors · 2019-11-14
> **Author Reply for Official Blind Review #2**
>
>
> First we want to thank reviewer #2 for the informative comments. We have updated the paper to incorporate the reviewer’s suggestions.
>
> Regarding point #1, we agree with the reviewer that current experiments are insufficient, and thus we have conducted additional experiments in the ant-direction mujoco environment used in other meta-RL papers. The results can be found at https://imgur.com/KPzosyd . Regarding the in-distribution performance difference between MIER and PEARL, we found it to be highly task-dependent. It is expected that in some in-distribution environments MIER would be slightly worse than PEARL because for a in-distribution test task, probabilistic inference is the optimal thing to do. However, we do want to make it clear that the focus of this paper is to improve the out-of-distribution task performance while maintaining similar performance on in-distribution tasks compared to existing methods. As suggested by the reviewer, we have also conducted additional experiments for out-of-distribution tasks in the humanoid direction environment, and the results can be found at https://imgur.com/Ob8K5Im . The x-axis corresponds to different test tasks, where task 0 is the easiest and -5 and 5 are the most out-of-distribution. We see that for almost all tasks, our methods outperforms PEARL. For all the additional experiments, we will incorporate the results in the final version of the paper.
>
> For point #2, we have modified the paper to include all the hyperparameter configurations in Appendix A.
>
> Regarding point #3, we want to thank the reviewers for pointing out the missing related works. We have modified the paper to include them in the related work section.
>
>
> Answers for questions:
> Q1: In the introduction: "Effective model training requires the validation batch to contain data corresponding to optimal behavior for the tasks...". Why? In principle we could train a good model of the environment by running a sufficiently-explorative policy.
>
> A1: We will modify the paper to make this more clear. We aim at suggesting that it is important to include data collected from the adapted policy in the validation batch, because the adapted policy might visit states that has never been visited by the unadapted policy. We agree with the reiwer that the same result could also be achieved with a sufficiently good exploration policy.
>
> Q2: In the related works: "Our method does not suffer from this problem since we use our model to train a model-free policy". It is not clear why (though it becomes later) since simulating long trajectories from a learned model could lead to the usual divergence issues.
>
> A2: Thanks for pointing this out. We have modified the paper to clarify this.
>
> Q3,4: In Sec. 3.2: r in \hat{p} should not be bold. Also, "f" in the subscript of the expectation was not defined (is it \hat{p}?). In Sec 3.4: there is a minimization over \phi which however does not appear in the objective.
>
> A3,4: These are indeed typos and we have fixed them. Thanks for pointing out.
>
>
> Q5: In the optimization problem at page 5: \phi_{\Tau} should probably be \phi.
>
> A5: It should be \phi_{\Tau}, since \phi_{\Tau} is the adapted context and it is a function of \phi. We are evaluating the final loss on the validation batch using adapted context.
>
> Q6: In Fig. 2, why is MIER run for less steps than the other algorithms?
>
> A6: We will re-run the experiments with more steps in the final version of the paper.

---

### Official Review · AnonReviewer3 · 2019-10-22
**Official Blind Review #3**

**Rating:** 3

**Review:**

###  Summary
1. The paper proposes an algorithm capable of off-policy meta-training (Similar to PEARL) as well as off-policy policy adaptation (By relabelling previous data using the adapted model and reward function).

2. The basic idea is to meta-learn a model that can adapt to different MDPs using a small amount of data. Moreover, the adaptation is done by only changing the latent context vector (Similar to CAVIA or CAML). The remaining parameters of the model (theta) are fixed after meta-training.

3. The paper also proposes learning a universal policy that, when given the context vector of a task, can maximize the reward for that task. This means that for with-in distribution meta-testing tasks, the policy can be used as it is (by giving it the right context vector which can be computed by adapting the model). For out-of-distribution tasks, however, it is important to update this policy.

4. To update the policy, the paper proposes combining previously stored data (for example data used in meta-training) with the adapted model to do off-policy learning (Using SAC).

### Decision with reasons

I vote for rejecting the paper in its current form for the following reasons:

1- The paper assumes that it is possible to learn models for out-of-distribution tasks with a few samples that are accurate on all the previously stored data. This is fundamentally incorrect. If the MDP changes in a significant way, it is not reasonable to expect that we can adapt a model from a few samples. Moreover, even if we can adapt the model using a lot of new experience, it is not reasonable to expect that we can use this model to accurately label all previous data. The authors do acknowledge this when describing results in Figure 3, however they seem to underplay this limitation.

2- Turning the meta-RL problem into a supervised learning problem has already been explored. For instance, Nagabandi et al. (2018)[1] showed that it is possible to quickly adapt models to changes using meta-learning. They, however, used decision time planning for the control policy (By random shooting method). This paper, on the other hand, uses Dyna style planning with an off-policy learning algorithm on previously stored data. The only difference is the choice of off-the-shelf planning algorithm which is not a significant contribution (There are some other small differences, such as learning a context vector and not model initialization, learning a universal policy etc, however, I don't see how they are essential for the proposed approach; maybe the authors can clarify why those choices are essential)

3- The paper assumes a context vector alone is sufficient to capture changes in MDPs (It keeps the rest of the model fixed at adaptation). This might be reasonable if the context vector is sufficiently large, but the paper does not even mention the size of the context vector. It also skips other important details. For example, it does not mention any details about hyper-parameter selection, how the context-vector used in the model, etc. It's hard to judge the importance of the experimental results because of this.

### Questions

1- "Effective model training requires the validation batch to contain data corresponding to optimal behavior for the tasks, which we obtain by training a universal policy conditioned on the context descriptor"

It's not clear to me why the validation batch must contain data corresponding to the optimal behavior.

2- Is the proposed framework really consistent? At adaptation, only the context vector is being updated whereas model parameters (theta) are fixed. Why is a context vector alone sufficient to adapt the model to drastic changes in the MDP?


[1] https://arxiv.org/abs/1812.07671

### UPDATE

The authors gave a detailed response to the reviews and answered some of my main concerns. However, I'm still not convinced that the paper, in its current form, can be accepted. My issues are:

The paper combines some existing ideas in a new way but falls short of justifying the choices it made. The proposed contribution is that it is consistent (meta-learning methods that learn a network initialization are also consistent), can do off-line meta-training (So can PEARL) and can use old meta-training data at meta-test time (This is novel to this paper). However,  the proposed methodology also has some downfalls. For example:

It does not allow continual adaptation. This is an important limitation of existing consistent meta-learning methods and this paper does not address it. Nagabandi et al 18 [1], on the other hand, propose a similar solution that is also capable of continual adaptation.

MOST IMPORTANTLY, the empirical evaluation in the paper is very unsatisfactory. Even though the authors have included hyper-parameters in the appendix in the updated version of the paper, they still do no specify how these parameters were selected. Were the parameters selected to maximize the performance of their method and then copied for the baselines? This would not be a fair comparison.

Given the above-mentioned issues, I don't think the paper in its current form can be accepted and I'm maintaining my initial score. I think the authors should do a more thorough empirical investigation and tune the baselines and their method separately (using comparable compute budget). They should also report results on multiple environments using the same parameters (i.e. tune hyper-parameters on one or a few environments and reports results on some other environments as commonly done in Atari)

[1] Deep Online Learning via Meta-Learning: Continual Adaptation for Model-Based RL

**Experience Assessment:**

I have published one or two papers in this area.

**Review Assessment: Checking Correctness Of Derivations And Theory:**

N/A

**Review Assessment: Checking Correctness Of Experiments:**

I carefully checked the experiments.

**Review Assessment: Thoroughness In Paper Reading:**

I read the paper thoroughly.

---

> ### Author Response · Authors · 2019-11-13
> **Author Reply for Official Blind Review #3**
>
> First we want to thank reviewer #3 for the constructive comments. We have updated the paper to incorporate the reviewer’s suggestions.
>
> Regarding point #1, as the reviewer points out, indeed it is impossible for the model to accurately adapt to a new task if the task is too out-of-distribution. We will modify the paper to make this clear. However, the main advantage of our algorithm is that the adaptation is consistent, meaning that given enough data during test time and a model with large enough capacity, the adaptation process would eventually perform well for the new task. This is crucial for adapting to out-of-distribution tasks, since there will be no guarantee on how the dynamics and reward function would change for the new tasks.
>
> Regarding point #2, as the reviewer points out, the proposed algorithm is not the first to formulate meta-RL problem into a meta-supervised learning problem. We agree with the reviewer that Nagabandi et al. (2018) should be included as a baseline. However, we weren’t able to reproduce the authors’ results using the open source code released by the authors, and we are actively communicating with them to resolve the problem. For now we ran our method on the HalfCheetah environment in Nagabandi et al. (2018), and the comparison can be found here: https://i.imgur.com/5bSCSgD.png. We see that our method achieves superior performance. We will modify the paper to include comparison to Nagabandi et al. (2018) in more environments once we resolve the problem.
>
> However, we do want to clarify that our proposed method is not merely a variation of planning methods on top of the model based approach described in Nagabandi et al. (2018). First of all, the off-policy relabeling method is not a planning algorithm, as it uses policy iteration to improve the policy instead of optimizing actions with respect to the dynamics and reward model’s prediction. Furthermore, the use of relabeling is essential for our method because it enables cross-task data reuse, which is a unique advantage only applicable in the meta-RL setting.
>
> As for point #3, as suggested by the reviewer, we have modified the paper to include all the hyperparameter configurations in Appendix A. We also want to make it clear that we are not making the assumption that the context vector alone is sufficient to capture changes in MDPs. On the contrary, we are assuming that the context vector is often insufficient to capture a new task, especially for out of distribution tasks. This is precisely the reason why we need off-policy data relabeling to continue improving the policy during test time. The ablation study on the right side of Figure 4 in our paper demonstrates the importance of relabeling using data from other tasks.
>
>
> Answers for questions:
> Q1: It's not clear to me why the validation batch must contain data corresponding to the optimal behavior.
> A1: We will modify the paper to make this more clear. We aim at suggesting that it is important to include data collected from the adapted policy in the validation batch, because the adapted policy might visit states that has never been visited by the unadapted policy.
>
>
> Q2: Is the proposed framework really consistent? At adaptation, only the context vector is being updated whereas model parameters (theta) are fixed. Why is a context vector alone sufficient to adapt the model to drastic changes in the MDP?
>
> A2: We want to clarify that by consistency, we mean that the proposed algorithm would converge to optimal policy asymptotically given enough data. As the reviewer points out, merely adapting the context would not guarantee consistency. However, as described in Algorithm 2 in the paper, during test time, we only adapt the context once but continue to adapt the dynamics and reward model by its full parameters. Therefore, if the model has large enough capacity to capture the ground truth dynamics and reward of the MDP, the continued adaptation is consistent.

---

> > ### Comment · AnonReviewer3 · 2019-11-14
> > **Acknowledging the response**
> >
> > Thank you for the detailed response.
> >
> > I've read it and I'm looking at the updated paper. I'll update my review soon (In ~6 hours)

---

> > ### Comment · AnonReviewer3 · 2019-11-15
> > **Another question**
> >
> > "However, as described in Algorithm 2 in the paper, during test time, we only adapt the context once but continue to adapt to the dynamics and reward model by its full parameters."
> >
> > If you update all the parameters of the model, wouldn't that destroy the meta-learned knowledge? i.e. the model wouldn't be able to 'quickly' adapt to further changes in the environment.

---

> > > ### Author Response · Authors · 2019-11-15
> > > **Author Reply for Official Blind Review #1**
> > >
> > > That's correct. Continuing to adapt the model would indeed prevent the model from rapidly adapting to other tasks. Therefore whenever we are testing on a new task, we always start from an unadapted model. However this is consistent with the standard meta-RL problem setup, where the model should not have seen any other test tasks during test time. A different problem setup which requires the model to be continually adaptable to different tasks is the online meta-learning problem [1], which we are not tackling in this paper.
> > >
> > > We also want to note that the continual adaptation approach that we use is common in other gradient based meta learning algorithms. For example, in the original MAML paper [2], the authors presented the continual adaptation results in Figure 3.
> > >
> > >
> > > [1] Finn, Chelsea, et al. "Online meta-learning." arXiv preprint arXiv:1902.08438 (2019).
> > >
> > > [2] Finn, Chelsea, Pieter Abbeel, and Sergey Levine. "Model-agnostic meta-learning for fast adaptation of deep networks." Proceedings of the 34th International Conference on Machine Learning-Volume 70. JMLR. org, 2017.

---

> > ### Comment · AnonReviewer3 · 2019-11-15
> > **Why modify a context vector as opposed to meta-learn model/policy initializations**
> >
> > What is the motivation behind meta-learning a context-vector based task identification mechanism as opposed to just meta-learning initialization as done by Nagabandi et al?
> >
> > I initially assumed the reason is that this mechanism would enable continuous adaptation of a single model  which model initialization doesn't (without storing a mixture over models, at least); however, after the authors clarified that they are, in fact, modifying all the parameters of the model at adaptation time, I don't see any benefit of context-vector based approach over what Nagabandi et al did. Am I missing something?

---

> > > ### Author Response · Authors · 2019-11-15
> > > **Author Reply for Official Blind Review #1**
> > >
> > > The goal of our paper is to develop a meta-RL method that is both consistent and sample-efficient. To achieve the sample efficiency, our method must be able to utilize off-policy data during training time. While we can indeed meta-learn the full initialization instead of a context vector for the reward and dynamics model using off-policy data, we cannot do the same for policy. Meta-learning the full initialization of the policy requires the use on-policy data during training time and therefore significantly sacrifice sample efficiency. Hence, in order to enable us to meta-train the policy with off-policy data, we need the adapted context vector from the reward and dynamics model to identify specific tasks.

---

### Decision · Program_Chairs · 2019-12-19

**Decision:**

Reject

**Comment:**

The authors propose an algorithm for meta-rl which reduces the problem to one of model identification. The main idea is to meta-train a fast-adapting model of the environment and a shared policy, both conditioned on task-specific context variables. At meta-testing, only the model is adapted using environment data, while the policy simply requires simulated experience. Finally, the authors show experimentally that this procedure better generalizes to out-of-distribution tasks than similar methods.

The reviewers agree that the paper has a few significant shortcomings. It's unclear how hyper-parameters are selected in the experimental section; the algorithm does not allow for continual adaptation; all policy learning is done through data relabelled by the model.

Overall, the problem the paper addresses is very important, but we do not deem the paper publishable in its current form.